# Bile Acids-Based Therapies for Primary Sclerosing Cholangitis: Current Landscape and Future Developments

**DOI:** 10.3390/cells13191650

**Published:** 2024-10-04

**Authors:** Stefano Fiorucci, Ginevra Urbani, Cristina Di Giorgio, Michele Biagioli, Eleonora Distrutti

**Affiliations:** 1Dipartimento di Medicina e Chirurgia, Università di Perugia, 06123 Perugia, Italy; urbaniginevra@gmail.com (G.U.); cristi.digiorgio@gmail.com (C.D.G.); michele.biagioli@unipg.it (M.B.); 2SC di Gastroenterologia ed Epatologia, Azienda Ospedaliera di Perugia, 06123 Perugia, Italy; eleonoradistrutti@gmail.com

**Keywords:** bile acids, biliary fibrosis, cholangiocytes, cholestasis, farnesoid X receptor, GPBAR1 (TGR5), microbiota, primary sclerosing cholangitis

## Abstract

Primary sclerosing cholangitis (PSC) is a rare, chronic liver disease with no approved therapies. The ursodeoxycholic acid (UDCA) has been widely used, although there is no evidence that the use of UDCA delays the time to liver transplant or increases survival. Several candidate drugs are currently being developed. The largest group of these new agents is represented by FXR agonists, including obeticholic acid, cilofexor, and tropifexor. Other agents that target bile acid metabolism are ASTB/IBAP inhibitors and fibroblasts growth factor (FGF)19 analogues. Cholangiocytes, the epithelial bile duct cells, play a role in PSC development. Recent studies have revealed that these cells undergo a downregulation of GPBAR1 (TGR5), a bile acid receptor involved in bicarbonate secretion and immune regulation. Additional agents under evaluation are PPARs (elafibranor and seladelpar), anti-itching agents such as MAS-related G-protein–coupled receptors antagonists, and anti-fibrotic and immunosuppressive agents. Drugs targeting gut bacteria and bile acid pathways are also under investigation, given the strong link between PSC and gut microbiota.

## 1. Introduction

Primary sclerosing cholangitis (PSC) is a chronic autoimmune liver disorder characterized by inflammation-driven immune dysfunction and fibrosis centered on bile ducts, leading to progressive cholestasis, biliary fibrosis, cirrhosis, and an increased risk of cholangiocarcinoma [1,2,3]. Pathologically, PSC presents with concentric and obliterative periductal fibrosis that affects both intrahepatic and extrahepatic bile ducts, known as “onion-skin” fibrosis, which leads to the formation of strictures within the biliary tree [1,4]. The infiltration of lymphocytes in the portal tract and subsequent inflammation drive the disease’s progression towards cirrhosis and cholangiocarcinoma (approx. 10–15% of PSC patients) [5]. PSC can affect both children and adults and is often linked with inflammatory bowel disease (IBD). PSC exhibits a strong correlation with inflammatory bowel disease (IBD), with approximately 80% of individuals diagnosed with PSC having either a prior history or concurrent manifestation (35–60%) of IBD, predominantly ulcerative colitis (UC), at the time of diagnosis [6,7]. Importantly, PSC concomitant with IBD constitutes a distinct clinical phenotype divergent from the isolated forms of either condition, and should be regarded as a composite disease characterized by dual-organ involvement. In contrast, it is estimated that about 5% of patients with IBD concurrently develop PSC. It is accepted that γ-glutamyltransferase (γGT) screening is both a cost-effective and accurate method for the early detection and diagnosis of PSC in IBD patients [3]. 

PSC is a rare disease. The incidence is 1–1.5 per 100,000 person-years and the prevalence is 6–16 per 100,000 person-years [8,9]. PSC associated with IBD is more common in males, with a 2:1 male-to-female ratio. Conversely, PSC presents with a slight female predominance in the absence of IBD. The average age at diagnosis typically falls between 30 and 40 years. As for primary biliary cholangitis (PBC), approx. 25% of PSC patients present with other autoimmune diseases, such as type I diabetes mellitus (T1DM), Hashimoto’s thyroiditis, celiac disease, psoriasis, and rheumatoid arthritis (Figure 1), and it is correlated to the HLA-DRB1*03031 haplotype and presents perinuclear anti-neutrophil cytoplasmic antibodies (pANCAs), considered as a prognostic marker for the disease [10]. 

In contrast to PBC, PSC typically involves large bile ducts and is better detected by magnetic resonance cholangiopancreatography (MRCP). Although there is reconsideration of the role of endoscopic retrograde cholangiopancreatography (ERCP) in the diagnosis of PSC, in the majority of centers, ERCP is currently performed only in patients that need therapeutic approaches such as bile duct dilation or stenting for acute bacterial cholangitis [11]. 

## 2. Clinical Variants of PSC

There are several clinical variants of PSC, which are shown in Table 1.

Clinically, the large majority of PSC patients develop cholestasis characterized by increased levels of alkaline phosphatase (ALP), γGT, and bilirubin, whose levels are progressive, but might also fluctuate over time. A minority of patients (<10%) have small-duct PSC, defined as a recent (<1 year) normal high-quality ERCP or MRCP, with histologic features typical of PSC or with both concurrent overt or histologic features of IBD and increased ALP levels (γGT in pediatric cases) [12].

Given the unknown etiology (Figure 1) [12], controversy exists as to whether PSC should be considered an autoimmune disease; in fact, despite the large number of autoantibodies detected in PSC patients, their specificity is generally low and their frequency is not sufficiently significant to be considered as diagnostic markers for the disease (in contrast to AMAs in PBC) [9], making MRCP the preferred diagnostic methodology for PSC [2]. The onset of PSC is usually poorly symptomatic, and symptoms develop slowly, which is why the large majority of patients are totally asymptomatic or have mild manifestations, such as fatigue, itching, and abdominal discomfort, at diagnosis. PSC is a progressive disease [13] with a significant risk of developing cirrhosis and cholangiocarcinoma, although no therapies have been shown to be effective in modifying the disease progression, and ≈40% of patients require liver transplantation at some point in their clinical history and develop end-stage liver disease [2].

## 3. Genetic and Immunopathogenesis of PSC

Despite the fact that the precise pathogenesis of PSC is currently unknown [14], the common association of PSC with IBD suggests that biliary inflammation develops as a consequence of liver influx of intestinal microbiota-derived antigens that promote liver inflammation and the recruitment of inflammatory leukocytes from the portal circulation [1] in genetically predisposed individuals.. This view is supported by early studies showing that, similarly to the inflamed intestine, inflamed sinusoidal cells in the liver express high levels of mucosal vascular address in cell adhesion molecule 1 or MAdCAM-1, which functions as an α4β7 integrin receptor and is critical for recruiting leukocytes to the intestinal mucosa during immune activation. Interaction of MAdCAM-1 with α4β7 integrin is an important target in IBD, and its blockade suppresses the development of liver fibrosis [15,16]. Activation of the liver immune system by intestinal antigens is associated with liver recruitment of multiple immune cell subsets, including macrophages and dendritic and NK cells, leading to NF-κB- and NLP3-dependent generation of cytokines and chemokines, including IL-6, IL-12, IL-1β, and TNF-α, which are instrumental for the further recruitment and activation of B and T lymphocytes (T_h_1 and T_h_17) that mediate biliary injury and maintain the disease [1,17]. Liver biopsies from PSC patients typically demonstrate a CD4^+^ T cell infiltrate that is prevalent in the portal area, while CD8^+^ T cells represent the majority of the lobular infiltrate [18]. 

PSC is considered a polygenic disorder. Genetics play a significant role in predisposing individuals to PSC, with both HLA and non-HLA genetic variants contributing to the risk. However, PSC is a complex disease, and its development likely involves a combination of genetic predisposition, immune dysregulation, and environmental factors. The strongest genetic association of PSC is found within the HLA complex, particularly in large-duct PSC (Table 1). Several other non-HLA loci have been identified in genome-wide association studies (GWAS) as being linked to PSC. These include: *MMEL1*, *TNFRSF14* [19,20], and *BCL2L11* [21]; *IL2*, *IL21*, *IL2RA*, *CD28*, *CTLA4*, *BACH2*, *SIK2*, *HDAC7*, *SH2B3*, *ATXN2*, *PRKD2*, *STRN4* and *PSMG1* [22]; *NFKB1*, *CCL20*, and *GPR35* [23,24,18]; *FOXP1*, *CCDC88B* [18], *MST1* [18], *RFX4*, *RIC8B*, *CLEC16A*, *SOCS1*, *TCF4*, *CD226*, and *UBASH3A* [24] (Figure 1). These genes are involved in immune regulation and cell signaling, including autophagy, senescence, and apoptosis and inflammation. Some of these genes are known predisposing factors for IBD, while there is only a limited number of specific genes that cause predisposition to non-IBD PSC.

## 4. Cholangiocytes Biology in PSC

Cholangiocytes, also known as biliary epithelial cells (BECs), are rare hepatic cells (~4% of the liver parenchyma) lining both intra- and extrahepatic biliary ducts [25]. Cholangiocytes’ main function is represented by primary bile (synthesized by hepatocytes) modification and transportation among the biliary tree via channels, transporters, and exchangers present on the apical or basal membrane [26,27]. Moreover, healthy BECs participate in homeostasis maintenance of the surrounding tissues via secretion of modulators involved in cell proliferation, senescence, and apoptosis [28], while, on the other hand, diseased cholangiocytes are responsible for biliary inflammation and fibrosis when they release pro-inflammatory and pro-fibrotic mediators such as tumor necrosis factor α (TNFα), interleukin 6 (IL-6), monocyte chemoattractant protein 1 (MCP-1), and transforming growth factor β (TGFβ) [29].

Precisely, two different types of BECs can be identified among the biliary tree: small ones, mainly located in small intrahepatic bile ducts, and large ones, mostly found in large intra- and extrahepatic bile ducts and the gallbladder [30,31,32]. Small flat or cuboidal cholangiocytes (diameter of ~5 µm) are quiescent cells characterized by a greater plasticity when compared to large columnar cholangiocytes (diameter of ~16 µm), as indicated by the higher nuclear-to-cytoplasmatic ratio: this is why small cholangiocytes, in response to various stimuli and/or insults, may proliferate and differentiate towards large cholangiocytes and might contribute to liver regeneration in the case of hepatocyte proliferation impairment, while large ones cannot [32,33,34]. On the apical plasma membrane, facing the bile duct lumen, both small and large BECs present numerous microvilli, which provide a five-fold increase in cell surface area, and a primary cilium acting as a mechano-, chemo-, and osmo-sensor for changes in bile flow and bile composition, which is responsible for stimulating fluid secretion or absorption from the cholangiocytes when needed [35,36,37]. In addition to their morphological differences, BECs demonstrate functional heterogeneity, too: differential protein expression between small and large cholangiocytes suggests that large, but not small, BECs are actively involved in bile modification [38,39].

In PSC, cholangiocytes of both intra- and extra-hepatic biliary ducts are affected by chronic and progressive inflammation, characterized by interlobular bile duct destruction associated with the typical concentric “onion skinning” fibrosis [40]. In cholangiopathies, such as PSC, cholangiocyte proliferation is fundamental to prevent ductopenia; however, persistent insult to cholangiocytes (e.g., microorganism infection, xenobiotics, environmental factors, endo- or exotoxins) leads to the development of the so-called ductular reactive cholangiocytes (DRCs), indicating the expansion of CK-7^+^ or CK-19^+^ cells with ductular reaction (DR) [41,42]. As was recently confirmed by both human and mouse model transcriptomic analyses, in chronic cholestasis, DRCs gain profibrotic and proinflammatory neutrophil-recruiting profiles characterized by the expression of mediators such as *IL-1β*, *IL-4*, *IL-6*, *iNOS*, *TNFα*, *MCP-1*, and *TGFβ* [43,44], which are responsible for increased portal infiltration by inflammatory macrophages and activated lymphocytes [45,46]. The correlation between the magnitude of the DRCs population and the fibrosis stage has allowed us to confirm cross-talk between DRCs and profibrogenic hepatic cells (i.e., HSCs, PFs), even though it is still not clear whether fibrosis is the cause or the result of activated cholangiocytes [47,48].

### 4.1. Senescence in PSC

PSC is increasingly considered as a disorder of accelerated cellular senescence of cholangiocytes, and it is now recognized that persistent insult can shift the cholangiocyte toward the senescence-associated secretory phenotype (SASP). In this setting, cholangiocytes, in addition to cell cycle arrest and apoptosis resistance, present increased β-galactosidase expression and impaired epithelial barrier function (cholangiocytes derived from PSC patients express display lower levels of ZO-1 tight junctions) [49] and secrete high levels of proinflammatory and profibrotic mediators, such as IL1β, L-6, IL-8, CCL2, MMP3, and PAI-1, which are not only responsible for sustaining the inflammatory and fibrotic response already in action, but also able to activate the senescent phenotype in surrounding, by-stander, and non-senescent cells [50,51] (Figure 2).

Notably, high-throughput next-generation sequencing (NGS) studies have identified a distinct pro-inflammatory senescence-associated secretory phenotype (SASP) profile in cholangiocytes isolated from PSC, signaled by the upregulated expression of genes such as *SERPINE1*, insulin growth factor-binding protein 5 (*IGFBP5*), and matrix metalloprotease 3 (*MMP3*), along with *IL-6* and *IL-8* and chemokines, such as *CCL2* [52,53]. Reduced telomerase reverse transcriptase (TERT) activity [54] has been observed in cholangiocytes from PSC patients, despite relatively preserved telomere length. These findings suggest that PSC pathogenesis may involve a stress-induced, telomere-independent senescence mechanism potentially driven by elevated expression of activated N-Ras, a potent promoter of cellular senescence. The presence of a substantial proportion of senescent cholangiocytes is clinically significant and correlates with disease severity; an increased burden of senescent cells is associated with more severe clinical outcomes [55].

### 4.2. Biliary Fibrosis

Biliary fibrosis, a hallmark of autoimmune cholangiopathies, is a key determinant of disease progression in PSC patients and is closely linked to worse clinical outcomes and higher liver histology scores. Within the liver, biliary fibrosis involves complex interactions between various cell types, particularly hepatic stellate cells (HSCs) and portal fibroblasts [56]. Although both HSCs and portal fibroblasts contribute to fibrosis, their roles and the timing of activation can differ significantly depending on the context and the model of biliary damage. The portal fibroblasts [57] are located in close proximity to bile ducts and are often the first responders to biliary injury. Upon activation, these cells proliferate and secrete ECM components, contributing to the initial stages of fibrosis. Their early response to biliary damage positions them as key players in the formation of peribiliary fibrosis, which can precede the development of more extensive fibrotic changes in peribiliary areas [26,58]. The HSCs, which, in the quiescent state, are located in the space of Disse, become activated in response to liver injury. In the context of biliary fibrosis, HSCs are often recruited later in the process, contributing to more advanced stages of fibrosis, such as bridging fibrosis, which links portal tracts and can progress to cirrhosis. Once activated, HSCs also produce ECM components and express contractile proteins that exacerbate the fibrotic response. Initiation of biliary fibrogenesis takes place in the portal, where cholangiocytes interact with HSCs, portal fibroblasts endothelial cells, and various immune cells (essentially macrophages). These cells communicate through a network of cytokines, growth factors, and extracellular matrix components, including IL-17 and CXCL8 [14], leading to the recruitment of portal myofibroblasts and HSCs, which, in turn, initiate and maintain the development of peri-biliary fibrosis (Figure 2).

## 5. Bile Acids Metabolism

Bile acids (BAs) are a broad class of steroids formed at the interface of the host and the intestinal microbiota. Chemically, there are two main groups of bile acids: primary and secondary bile acids, based on their origin and metabolic pathway [59]. Primary bile acids are synthesized in the liver from cholesterol through enzymatic processes initiated by cholesterol-7α-hydroxylase (CYP7A1) or sterol 27-hydroxylase (CYP27A1) [60,61]. These pathways lead to the production of chenodeoxycholic acid (CDCA) and cholic acid (CA). In hepatocytes, CA and CDCA undergo conjugation with glycine or taurine, creating bile salts. This conjugation, catalyzed by bile acid-CoA synthetases (BACS) and bile acid-CoA: amino acid N-acyltransferases (BAAT), reduces the pKa of bile acids, increasing their solubility and facilitating their secretion into the bile [62]. Additionally, bile acids can be conjugated with glucuronic acid or sulfated by various enzymes, which are present in larger quantities in conditions like PBC, PSC, and pregnancy-related cholestasis [41].

Conjugated bile acids are actively secreted across the hepatocyte’s biliary membrane into the canalicular space via the bile salt export pump (BSEP), initiating bile flow [63]. While in fasting, bile acids are stored in the gallbladder, they are the released into the intestine in response to feeding. In the ileum and colon, primary bile acids undergo further transformations mediated by the gut microbiota. These transformations include deconjugation, 7-dehydroxylation, hydroxylation, oxidation, and epimerization, producing secondary bile acids like lithocholic acid (LCA) and deoxycholic acid (DCA) [64,65,66]. Bacterial enzymes such as bile salt hydrolases (BSHs) play a critical role in these processes. Following deconjugation, 7α-dehydroxylation generates LCA and DCA, primarily mediated by bacteria like *Clostridium scindens* and *Clostridium hiranonis*. Hydroxysteroid dehydrogenases (HSDHs) further modify bile acids and these 3-oxo derivatives [67].

### 5.1. Bile Acid-Regulated Receptors (BARRs)

Bile acids regulate various physiological processes, including lipid metabolism, energy balance, and immune responses, through interactions with membrane and nuclear receptors, collectively known as bile acid-regulated receptors (BARRs) [68]. Several of these receptors play a role in PSC.

#### 5.1.1. Membrane Receptors

GPBAR1 is a G-protein-coupled receptor activated by secondary bile acids (DCA, LCA) and their derivatives. It is primarily expressed in the ileum, colon, spleen, adipose tissue, and lymphoid tissues [69]. GPBAR1 modulates immune responses, particularly in myeloid cells such as monocytes and macrophages, by inhibiting pro-inflammatory pathways like NF-κB signaling and NLRP3 inflammasome activation [70]. GPBAR1 exerts a major role in regulating cholangiocyte homeostasis and is considered an important target in PSC [71]; it will be reviewed below. Various GPBAR1 agonists are currently under development [72,73] for treating liver and intestinal disorders [74].

##### S1P2R

This receptor, activated by sphingosine-1-phosphate (S1P) and conjugated bile acids like TCA, is expressed in the liver and immune cells. S1P2R promotes inflammatory responses and fibrosis in the liver, making it a potential target in liver diseases [75].

##### Leukemia Inhibitory Factor Receptor (LIFR)

LIFR is involved in immune regulation and is activated by leukemia inhibitory factor (LIF). Bile acids such as glycodeoxycholic acid (GDCA) can antagonize LIFR, potentially providing therapeutic benefits in LIF-expressing cancers and immune regulation [76,77].

#### 5.1.2. Nuclear Receptors

The farnesoid X receptor (FXR) is a key regulator of bile acid metabolism, particularly through its effects on CYP7A1 and CYP8B1 expression [78]. FXR activation [79] leads to the repression of these genes via *SHP* (small heterodimer partner) [80], reducing bile acid synthesis. FXR also plays an important role in immune regulation by inhibiting pro-inflammatory genes and the *NLRP3* inflammasome [81].

FXR plays a major role in regulating bile acid homeostasis. As described in Figure 3, FXR is essential for regulating bile acid absorption, synthesis, and excretion by hepatocytes, regulating bile acid homeostasis and reducing intracellular bile acid accumulation through a multi-layered mechanism: (a) repression of bile acid import transporters: FXR reduces the expression of bile acid transporters such as NTCP (sodium/taurocholate co-transporting polypeptide) [82] in the liver and ASBT (apical sodium-dependent bile acid transporter in the ileum), limiting bile acid uptake into cells [83]; (b) induction of bile acid export pumps in the liver and intestine: FXR enhances the activity of transporters like BSEP (bile salt export pump), MRP2 (multidrug resistance protein 2), and OSTα/β (organic solute transporter α/β), promoting the excretion of bile acids from hepatocytes and ileal cells [84]; (c) suppression of bile acid synthesis: FXR activation downregulates CYP7A1, both directly in hepatocytes via SHP (small heterodimer partner)-induced repression and indirectly through intestinal FGF19 signaling, which also represses CYP7A1 in the liver [85]; (d) regulation of bile acid composition: FXR controls CYP8B1, another enzyme in bile acid synthesis, influencing the ratio of CA/CDCA. Unlike CYP7A1, FXR downregulates CYP8B1 through mechanisms involving SHP and the transcriptional repressor MAFG, thereby reducing CA production [86]; (e) detoxification and excretion: FXR promotes bile acid detoxification by inducing enzymes like CYP3A4, SULT2A1, and UGT2B4, which facilitate bile acid metabolism. It also enhances the secretion of biliary phospholipids through MDR3 (ABCB4) [87]. In contrast, it has been suggested that FXR might function as negative regulator of MRP4 [88], which supports the concept that FXR antagonism (rather than its agonism) might be beneficial in cholestasis [89] by promoting the release of bile acids from hepatocytes into the systemic circulation [90]. It has to be noted, however, that despite these protective effects, pharmacological activation of FXR by obeticholic acid (OCA) might increase cholesterol saturation and bile acid hydrophobicity in bile, potentially elevating the risk of gallstone formation [91]. This paradoxical effect suggests that FXR activation can have complex impacts on cholesterol metabolism.

In the small intestine, FXR regulates bile acid absorption and lipid metabolism by controlling the expression of key transporters, including the apical sodium-dependent bile acid transporter (*ASBT*) and organic solute transporters α and β (*OSTα*/*OSTβ*). FXR negatively regulates *Asbt* expression in mice, indicating species-specific differences in bile acid regulation. Inhibition of *ASBT* can lead to significant bile acid malabsorption, suggesting that ASBT inhibitors may have therapeutic potential in treating cholestatic disorders. The Fibroblast growth factor 19 (FGF19) [92], secreted in the small intestine upon FXR activation, plays a key role in regulating bile acid synthesis in the liver by binding to FGFR4 and β-Klotho. This interaction inhibits the expression of CYP7A1 through the activation of the JNK pathway, reducing bile acid production [92,93]. FGF15/19 also regulates gallbladder filling by interacting with FGFR3 [94].

Additionally, FXR exerts a variety of immune-regulatory functions [95,96,97], reviewed elsewhere [68], that could be beneficial in cholestatic disorders.

Pregnane X Receptor (PXR) responds to toxic bile acids like LCA and regulates bile acid detoxification pathways [98]. PXR also modulates immune responses and inflammation by interacting with NF-κB and inflammasomes [99].

Vitamin D Receptor (VDR) responds to LCA [100] and mediates some of the bile acids’ anti-inflammatory effects, contributing to bile acid regulation and immune modulation.

Retinoid orphan receptor gamma t (RORγt): Several secondary bile acids, including 3-oxo-DCA and Iso-Allo-LCA function as inverse agonists of RORγt, a transcription factor involved in the regulation of immune responses, particularly through its role in the differentiation of T_h_17 cells and the production of pro-inflammatory [101,102,103]. RORγt exerts a dual role in both protecting the body from infections and contributing to autoimmune and inflammatory diseases when dysregulated. Its involvement in gut homeostasis, autoimmune diseases, and cancer makes it an important target for therapeutic intervention [104]. RORγt is essential for: (1) the differentiation of CD4^+^ T helper cells into T_h_17 cells, a subset of T cells that produce pro-inflammatory cytokines such as IL-17, IL-21, and IL-22. T_h_17 cells are involved in various process including in autoimmune diseases such as multiple sclerosis, rheumatoid arthritis, psoriasis, and inflammatory bowel disease. However, it should be considered that T_h_17 cells protect against extracellular bacterial and fungal infections, particularly at mucosal surfaces. (2) Regulation of inflammation. RORγt’s role in promoting the secretion of IL-17 makes it a central regulator of inflammation, particularly in autoimmune and inflammatory diseases. IL-17 is a potent pro-inflammatory cytokine, and overexpression or dysregulation of RORγt can lead to chronic inflammation and tissue damage in autoimmune conditions. (3) RORγt is also critical for the development and function of type 3 innate lymphoid cells (ILC3s) [105], which are important for maintaining the balance of gut immunity and the microbiome. ILC3s produce IL-17 and IL-22, cytokines involved in protecting mucosal barriers in the gut and other tissues. These cells are key in defending against pathogens while also regulating immune tolerance to commensal bacteria in the gastrointestinal tract. (4) Involvement in maintaining intestinal immune homeostasis: RORγt controls the balance between pro-inflammatory and anti-inflammatory signals in the gut [68]. Dysfunction of RORγt can lead to gut-related conditions, including IBD [105]. T_h_17 cells and ILC3s, driven by RORγt, contribute to the inflammatory processes in Crohn’s disease and ulcerative colitis/ (5) Suppression of regulatory T cells (Tregs) [106]. RORγt often functions in opposition to Foxp3, the master regulator of regulatory T cells (Tregs). While Tregs are involved in maintaining immune tolerance and preventing autoimmunity, RORγt promotes immune activation through T_h_17 cells. An imbalance between these two can lead to immune dysregulation. (6) RORγt has a role in anti-tumor immunity, particularly through its effects on T_h_17 cells and IL-17 production. However, its role in cancer is complex, exerting both growth and suppression effects on solid cancers.

### 5.2. Bile Acids in PSC

Numerous studies have investigated bile acid composition in PSC and other cholangiopathies. The majority of these have concentrated on serum bile acid composition as a reflection of the bile acid pool within enterohepatic circulation. Generally, these findings indicate that patients with PSC exhibit elevated serum levels of total, primary (CA and CDCA), and conjugated bile acids, along with an increased primary-to-secondary bile acid ratio. Conversely, fecal bile acid levels were found to be lower in PSC patients with IBD compared to those without PSC. An earlier investigation into both PBC and PSC groups, assessing 17 bile acids in serum, revealed heightened levels of total bile acids as well as taurine and glycine conjugates of primary bile acids in both cohorts. In PSC patients, secondary bile acids were notably diminished. More recent research on mostly non-cirrhotic individuals found that most bile acids in bile fluid were reduced in PSC, with the exception of TLCA, a harmful secondary bile acid, supporting earlier findings [25,26,27,28,29].

More recently, bile acid synthesis in PSC patients has been assessed by measuring the levels of the 7α-hydroxy-4-cholesten-3-one (C4) as a surrogate biomarker for bile acid synthesis [107]. The study demonstrated the prognostic significance of C4 levels in patients with PSC. Specifically, it was found that lower C4 levels, indicating suppressed bile acid synthesis, are associated with worse clinical outcomes, including an increased risk of liver transplantation or death. Lower C4 levels were negatively correlated with liver transplantation-free survival in PSC patients, and treatment UDCA, a common treatment for PSC, did not appear to influence bile acid synthesis or C4 levels, indicating that its therapeutic effects might be independent of bile acid synthesis modulation. These findings suggest that excessive suppression of bile acid synthesis is linked to poorer outcomes, highlighting the risk that in patients with advanced PSC, where bile acid synthesis is already suppressed, further reduction may be harmful. These results also suggest that therapeutic strategies targeting bile acid synthesis by targeting the FXR–FGF19 axis should carefully consider and take into account the extent of bile acid synthesis suppression and avoid further suppression in patients with advanced PSC. The study provides crucial insights into the role of bile acid synthesis in PSC and highlights the potential risks of targeting this pathway in patients with advanced disease. These findings emphasize the need for more nuanced therapeutic strategies and careful patient selection in clinical trials to improve outcomes in PSC.

In summary, bile acids and their regulated receptors play a critical role in PSC, influencing bile acid metabolism, transport, and immune responses. Their interactions with gut microbiota and various receptors offer potential therapeutic targets in the cholestatic liver.

## 6. Intestinal Microbiota in PSC

Intestinal dysbiosis has been consistently documented in PSC patients, regardless of the presence of IBD. The microbiota in PSC patients is characterized by an increased abundance of specific microbial taxa, including *Enterococcus* spp. (particularly *Enterococcus faecalis*), *Fusobacterium*, *Veillonella* spp., and *Enterobacteriaceae* (e.g., *Escherichia coli* and *Klebsiella pneumoniae*), alongside *Lactobacillus* spp. and *Streptococcus* spp. Conversely, there is a notable decrease in the abundance of *Coprococcus* spp. and *Blautia* spp. [108,109,110]. These microbial shifts are reminiscent of changes observed in Abcb4-/-mice models [111], suggesting that the intestinal microbiota in PSC patients deviates substantially from that of healthy controls. This dysbiosis is marked by an increase in potentially pathogenic species and a reduction in beneficial bacteria.

Despite these observations, there remains a lack of consistency in pinpointing specific microbial species and elucidating the precise mechanisms linking intestinal dysbiosis to liver immune system activation. Nonetheless, the modulation of the intestinal microbiota is emerging as a promising area of research in PSC. This notion is supported by evidence indicating that long-term antibiotic therapy with metronidazole and vancomycin can positively impact biochemical parameters, Mayo risk scores, and clinical symptoms in PSC patients. However, while these findings support a mechanistic relationship between PSC and gut microbial dysbiosis, there are currently no data demonstrating the effect of microbiome-based therapies on disease progression [58,112].

The mechanistic involvement of the gut in PSC remains ambiguous. Early theories proposed a “leaky gut” mechanism, where bacterial components and products might migrate passively through the portal circulation to induce biliary inflammation. However, it remains unclear whether alterations in the gut microbiota are causative of PSC development or represent secondary changes due to disease progression, or whether both scenarios occur simultaneously. The gut microbiota undoubtedly plays a significant role in bile acid homeostasis and may influence bile physiology either directly or indirectly, possibly through altered FXR and GPBAR1 signaling.

## 7. Assessing Therapeutic Efficacy in PSC

### 7.1. Biochemical Biomarkers

Various predictive models based on clinical and laboratory data have been developed to gain prognostic information on PSC management [2]. The Revised Mayo Risk Score is the most extensively validated model, particularly for short-term prognostication; however, it has proven inadequate for broader clinical trial applications, especially beyond mortality endpoints, and does not account for variant PSC phenotypes. Current models, including the Amsterdam—Oxford, UK-PSC [113], Sclerosing Cholangitis Outcomes in Pediatrics (SCOPE), and PReSTo score, represent substantial advancements, offering superior predictive accuracy for both short-term and long-term outcomes. These models have consistently outperformed the Revised Mayo Risk Score, with specific applicability to distinct patient subgroups, such as pediatric populations (SCOPE), patients with overlap PSC and autoimmune hepatitis (Amsterdam—Oxford), and those with small-duct PSC (UK-PSC and Amsterdam—Oxford).

The PReSTo model, developed using machine learning algorithms, consists of nine variables [114]: bilirubin, albumin, serum alkaline phosphatase (SAP) times the upper limit of normal (ULN), platelets, AST, hemoglobin, sodium, patient age, and the number of years since PSC was diagnosed. Validation in an independent cohort confirmed that PREsTo accurately predicts decompensation, and it performed well compared to MELD score and Mayo PSC risk score. The application of these prognostic tools should be carefully tailored to the disease stage, patient cohort, and intended outcome measures, such as transplant-free survival versus the onset of decompensated cirrhosis. This stratified approach is particularly relevant for therapeutic trials, where these models can identify patients most likely to respond to specific mechanisms.

### 7.2. Radiology

The prognostic utility of imaging-based scoring systems in PSC is undergoing a resurgence of interest. Historically, the Amsterdam criteria, derived from endoscopic retrograde cholangiopancreatography (ERCP) imaging, demonstrated significant prognostic value; however, the transition to magnetic resonance cholangiopancreatography (MRCP) as the preferred diagnostic modality diminished its clinical application. The ANALI score [115], which integrates the extent of biliary stricture burden with hepatic parenchymal abnormalities—including hepatic dysmorphia and radiological signs of portal hypertension on MRI—has shown robust predictive capacity for hepatic decompensation and survival without liver transplantation. There is potential to further refine prognostication by combining the ANALI score with the liver stiffness measurement (LSM) [116]. A recently developed imaging-based tool, the DiStrict score [117], has been proposed for the stratification of patients with large-duct PSC. The score was developed based on the extent and severity of cholangiographic changes in intrahepatic and extrahepatic bile ducts on 3D-MRCP and is based on the presence and extent of biliary strictures and dilatations. The DiStrict score seems able to predict liver transplantation- and liver-related death and patient survival, but was developed using a retrospective study and needs prospective validation [117].

### 7.3. Staging of Liver Fibrosis

Non-invasive assessment of liver fibrosis is strongly recommended in the management of PSC and can be conducted using liver stiffness measurement (LSM) techniques such as transient elastography (TE) and magnetic resonance elastography (MRE) [4]. These techniques are valuable for determining the stage and progression of fibrosis, with TE showing a diagnostic threshold of 14.4 kPa for stage 4 fibrosis. An annual increase in stiffness greater than 1.3 kPa has been associated with reduced transplant-free survival rates. However, TE is limited in PSC by transient biliary obstructions and the heterogeneous distribution of biliary strictures, which can lead to inaccurate readings. Magnetic resonance elastography (MRE) has been shown to be effective in identifying patients at risk for hepatic decompensation, with a stiffness threshold of 4.32 kPa. Despite the effectiveness of these techniques in risk stratification, no therapeutic clinical trial has yet demonstrated a reduction in fibrosis as measured by LSM in PSC patients. A phase 2b clinical trial of simtuzumab involving paired liver biopsies and ELF scores obtained 96 weeks apart demonstrated that baseline ELF scores were predictive of fibrosis progression by one or more stages [118]. Other biochemical scores, such as the Fib-4, have shown limited diagnostic accuracy when compared to MRE [119].

## 8. Treatment of PSC

The lack of effective therapeutic options for PSC constitutes a critical and glaring unmet need (Figure 4), especially when compared to the considerable progress achieved in the treatment of other chronic liver diseases in the past [1,120]. Some of the primary challenges of drug development in PSC patients are, in addition to the low prevalence of the disease, the relatively slow rate of fibrosis progression and the infrequency of clinical events [13], which complicate the evaluation of potential therapeutic agents. Given the similarity with PBC, the large majority of therapeutic attempts have been focused on bile acids. These approaches include UDCA and UDCA-related agents such as NorUDCA, as well as direct FXR agonists, such as OCA and cilofexor and tropifexor, or FXR-regulated agents such as FGF19 analogues or ASTB-IBAP inhibitors (Figure 4). In addition, similarly to PBC, peroxisome proliferator-activated receptor (PPAR) agonists, such as elafibranor, are currently under evaluation in clinical trials (Figure 4). Ongoing clinical trials on novel PSC therapeutics are reported in Table 2.

### 8.1. UDCA in PSC

While UDCA (15 mg/kg/day) is considered the first-line treatment for PBC, its effectiveness in PSC patients is less clear [9]. A randomized, placebo-controlled trial involving 105 PSC patients [121] who received 13 to 15 mg/kg/day of UDCA or a placebo, carried out at the Majo Clinic, Rochester, USA [122], showed a significant reduction in ALP levels, although there were no significant effects on histological progression or the time to liver transplantation. Similar results were reported in 2005 by Olsson et al., who enrolled 219 PSC patients in a randomized placebo-controlled study testing UDCA at a dose of 17–23 mg/kg body weight or placebo [123]. Results were available for 198 participants and demonstrated a trend toward improved survival or progression to CCA in PSC patients administered UDCA, but the results were not statistically significant even at these higher doses. However, the study was underpowered to detect a 50% reduction in adverse outcomes, taking into consideration the low incidence of death and liver transplantation in PSC patients.

The effects of higher doses of UDCA (28 to 30 mg/kg/day) were assessed in a long-term, 5-year, randomized, double-blind, controlled trial by Lindor et al. [124]. Although this higher dosage significantly lowered serum ALP and ALT levels, Lindor et al. concluded that “at the end of the study, 30 patients in the UDCA group (39%) versus 19 patients in the placebo group (26%) had reached one of the pre-established clinical endpoints. After adjustment for baseline stratification characteristics, the risk of a primary endpoint was 2.3 times greater for patients on UDCA than for those on placebo (*p* < 0.01) and 2.1 times greater for death, transplantation, or minimal listing criteria (*p* = 0.038). Serious adverse events were more common in the UDCA than placebo group (63% vs. 37%: *p* < 0.01)” [124]. This unexpected increase in adverse events in PSC exposed to UDCA has been highly debated over the years, raising concerns about the potential risks of high-dose UDCA in PSC.

In pediatric patients, a retrospective study including 287 children [125] reported that treatment with UDCA at the mean dose of 17 mg/kg/day was associated with a higher reduction in γGT levels after 1 year of treatment, while a reduction in γGT levels below 50 IU regardless of the use of UDCA was associated with improved outcomes. Given that low to moderate doses of UDCA (13–23 mg/kg/day) are considered safe and that significant reductions in ALP correlate with better outcomes, the most recent guidance from the American Association for the Study of Liver Diseases (AASLD) suggests that UDCA may be offered [33].

### 8.2. UDCA Derivatives

24-Norursodeoxycholic acid (NorUDCA) is a modified version of UDCA with a shorter side chain that undergoes intrahepatic recycling, a process known as cholehepatic shunting, which stimulates bicarbonate-rich bile [126,127,128]. NorUDCA has demonstrated anti-inflammatory, anti-fibrotic, and anti-proliferative effects in preclinical models of PSC. It has also been shown to lower serum levels of ALP, γGT, AST, and ALT in phase 2 trials involving PBC patients, with a phase 3 trial currently underway to assess its impact on histological progression (NCT03872921)) [9]. In Abcb4-/-mice, a genetic PSC model, NorUDCA, exhibited immunomodulatory properties by reducing liver levels of innate and adaptive immune cells, including CD8^+^ leukocytes [129]. In a non-cholestatic model of CD8^+^ T cell-driven immunopathology, triggered by acute LCMV infection, NorUDCA lessened liver damage and systemic inflammation. Mechanistically, it reduced CD8+ T cell infiltration by inhibiting lymphoblast formation, glycolysis, and mTORC1 signaling [129]. These findings suggest that NorUDCA directly modulates CD8^+^ T cells and mitigates CD8^+^ T cell-driven liver damage through a mechanism that is independent of FXR and GPBAR1.

### 8.3. FXR Agonists in PSC

As reported above, OCA is a potent FXR agonist that attenuates inflammation and liver fibrosis [130]. OCA has been approved as a second-line add-on therapy to UDCA in PBC [131] due to evidence of biochemical efficacy in a phase III study. In addition, current evidence has demonstrated that OCA increases transplant-free survival in PBC patients, despite the risk of major side effects [132]. In PSC patients, 5–10 mg OCA once a day for 24 weeks was found to be partially effective in reducing ALP levels (NCT02177136; EudraCT: 2014-002205-38), with no change in fibrosis markers [133]. Pruritus occurred in 67% of patients taking 5–10 mg/day and was dose-dependent [133]. However, the use OCA in PBC patients has been plagued by several side effects. These adverse events can vary in severity, with common side effects being mild and others potentially more serious. Common adverse events associated with the use of OCA include pruritus [134], which is dose-dependent, occurs in a significant proportion of patients, and can be severe in some cases, sometimes requiring dose adjustment or discontinuation. OCA use might increase LDL cholesterol levels [135], which could heighten the risk of cardiovascular events. This may require the use of cholesterol-lowering medications in some patients. Serious adverse events include the worsening of liver function in some cases, particularly in those with cirrhosis or advanced liver disease, leading to liver failure and, in rare cases, death. Hepatic decompensation has been reported in patients with advanced cirrhosis. The severity of these side effects will likely preclude the further development of OCA in PSC. Furthermore, the European Medicines Agency has recently recommended the revoking of conditional marketing authorization of OCA in PBC treatment (https://www.ema.europa.eu/en/news/ema-recommends-revoking-conditional-marketing-authorisation-ocaliva) (28 June 2024).

Cilofexor, a nonsteroidal FXR agonist, has been also tested in phase 2 trials in PBC patients, while a phase III clinical trial, the PRIMIS study, has been discontinued (ClinicalTrials.gov NCT03890120 registered 26 March 2019). In a 12-week phase 2 randomized controlled trial, cilofexor achieved a 21% reduction in ALP levels [136]. However, the subsequent phase 3 trial (PRIMIS), which involved 96 weeks of therapy and included paired biopsy specimens from patients with non-cirrhotic large-duct PSC, was terminated earlier because the interim futility analysis showed that the estimated probability of meeting the primary endpoint was 6.8%. At week 96, the proportion of patients with a ≥ 1 stage increase in fibrosis (Batts Ludwig stage) was 30.8% in the cilofexor group compared with 32.8% in the placebo group.

Tropifexor, another FXR agonist, has shown some efficacy in liver steatosis [137] and in PBC patients in a phase II trial [138], but has never been investigated in PSC.

The future role of FXR agonists in PSC therapy may hinge on the use of combination therapies that target multiple mechanisms of action, given the complexity and multifactorial nature of the disease. This approach could potentially enhance the therapeutic efficacy of FXR agonists and address the limitations observed in monotherapy trials [139].

### 8.4. FGF-19

FGF-19 is a hormone produced in the liver, gallbladder, and distal small intestine upon activation of FXR [140]. FGF-19 [141] plays a crucial role in inhibiting bile acid synthesis by downregulating CYP7A1 after binding to its receptor, the FGFR4 [142]. However, it is important to note that FGF-19 can also stimulate cell proliferation in the liver and intestine via IL6/STAT3 signaling pathways, which have been implicated in the development of hepatocellular carcinoma in murine models [143]. NGM282, Aldafermin [144], is a synthetic analogue of FGF-19 engineered to potentially benefit cholestatic liver diseases by prolonging the inhibition of bile acid synthesis. This analogue has been modified to achieve biased FGFR4 signaling, thereby preserving FGF-19′s regulatory effects on bile acid metabolism while avoiding the activation of STAT3 signaling and reducing the risk of tumorigenesis. In a phase II randomized controlled trial involving 62 patients with PSC [145], NGM282 did not achieve the primary endpoint of significantly reducing ALP levels at 12 weeks, but improved liver fibrosis, as measured by the ELF test and pro-collagen 3, a marker of fibrogenesis. Additionally, reductions were observed in C4, a marker of bile acid synthesis, and total bile acids. However, there are several oncogenic concerns over the development of the FGF19 analogue in general, since FGFR4/βklotho signaling in hepatocytes contributes to cancer cell survival by promoting the expression of PDL1 [146,147]. The oncogenic potential of FGF19 analogue should be carefully weighted, especially in patients with PSC, who are at increased risk of developing cholangiocarcinoma.

In general, however, the interplay between FXR/FGF-19 signaling and changes in the gut microbiome in PSC is a topic that warrants further investigation, as it could lead to new therapeutic strategies. This is particularly relevant given the close anatomical and functional relationship between inflammatory processes and the enterohepatic circulation of bile acids in PSC [143,148].

### 8.5. GPBAR1 in PSC

As previously discussed, GPBAR1 (TGR5) is viewed as a promising therapeutic target for managing cholestatic diseases due to its anti-inflammatory, anti-apoptotic, choleretic, and bicarbonate-dependent cell-protective mechanisms [149,150]. Although GPBAR1-deficient mice do not naturally develop cholestatic disorders, research has shown that damage to cholangiocytes and inflammation in PSC patients is linked to a significant reduction in the expression and function of GPBAR1 in biliary epithelial cells (BECs) [70]. A similar pattern is observed in BECs isolated from Abcb4-/-mice [149]. However, treatment of PSC-derived BECs with UDCA or NorUDCA has been shown to counteract the downregulation of GPBAR1 in these cells. Given that UDCA activates GPBAR1 in myeloid cells [151] and has anti-inflammatory effects in IBD patients, as well as modulating the gut microbiota in IBD models, it is speculated that selective activation of GPBAR1 could be beneficial in treating PSC. Supporting this hypothesis, genome-wide association studies have identified mutations in the GPBAR1 gene (located on chromosome 2q35) in PSC patients [152]. Sequencing of 274 healthy controls and 267 PSC patients revealed six nonsynonymous mutations and 16 novel single-nucleotide polymorphisms. Notably, five of the six nonsynonymous mutations were found to impair or eliminate GPBAR1 function. Despite these promising findings, there are currently no GPBAR1-targeted therapies available for PSC.

### 8.6. ASBT/IBAT Inhibitors

The apical sodium-dependent bile acid transporter (ASBT) known as ileal bile acid transporter (IBAT), encoded in humans by the SLC10A2 gene, is crucial for the enterohepatic recirculation of bile acids, facilitating their reabsorption in the terminal ileum [144]. FXR activation downregulates ASBT, thereby diminishing bile acid circulation and reducing the bile acid pool. This pathway has been implicated in ameliorating cholestatic liver conditions in animal models, suggesting potential therapeutic value for ASBT inhibitors in PSC [144]. Results from an open-label Phase II trial conducted to assess the efficacy of LUM001 (maralixibat), an IBAT inhibitor [153], in 27 PSC patients revealed no significant alterations in liver biochemistry endpoints, while serum bile acids were reduced. Other IBAT inhibitors were Volixibat and Odevixibat (NCT04663308, NCT02061540, NCT05642468).

### 8.7. PPARs

Fibrates, commonly used in the treatment of hypertriglyceridemia, act as PPAR-α and PPAR-δ agonists, exerting anti-inflammatory and metabolic effects. Fenofibrate has recently become an off-label therapeutic option for patients with UDCA refractory cholestasis [9] and exerts anti-inflammatory effects [154]. Bezafibrate, a pan–peroxisome proliferator-activated receptor (PPAR) agonist, has shown promise in treating primary biliary cholangitis (PBC) in patients who exhibit an inadequate response to ursodeoxycholic acid (UDCA), although it has not yet gained clinical approval in the United States. Fibrates, as a class, have demonstrated the ability to significantly reduce bile toxicity and enhance bile acid glucuronidation in patients with PSC when combined with UDCA therapy. Several small retrospective and prospective studies have indicated that fenofibrate and bezafibrate can improve cholestasis in PSC patients. Notably, a recent randomized, double-blind, placebo-controlled trial involving PSC patients reported a 66% reduction in ALP levels with Fenofibrate treatment, compared to a 20% reduction in the placebo group. This underscores the potential of fibrates as an adjunct therapy in PSC management.

Currently, a French multicenter study is investigating the effects of bezafibrate over a 24-month period in PSC patients with persistent cholestasis despite UDCA treatment (NCT04309773). Furthermore, studies exploring novel fibrates, such as elafibranor, are also in the pipeline (NCT05627362). These ongoing trials may pave the way for new therapeutic strategies in PSC, offering hope for more effective management of this challenging disease.

### 8.8. Antifibrotic Agents

The fibrotic characteristics of biliary strictures in PSC underscore the necessity for continued exploration of antifibrotic therapies. Cenicriviroc [155], a dual antagonist of CCR2 and CCR5 anti-fibrotic, was originally tested in an open-label, single-arm study for its efficacy in reducing ALP in PSC patients with minimal efficacy [156]. Since the development of this agent has now been abandoned for lack of efficacy, no further data are available. Simtuzumab, an inhibitor of Lysyl Oxidase-Like 2 (LOXL2), initially demonstrated potential in preclinical and early clinical studies; however, a subsequent Phase 2 clinical trial failed to show a significant therapeutic effect [118]. Emerging antifibrotic agents include bexotegrast (PLN-74809), which targets integrins αVβ6 and αVβ1 to inhibit the activation of transforming growth factor-β (TGF-β). This compound is currently under investigation in idiopathic pulmonary fibrosis and is also being evaluated for its efficacy in PSC in a Phase 2 study (NCT04480840).

### 8.9. Immunosuppression

Vidofludimus calcium functions as a dihydroorotate dehydrogenase inhibitor, promoting apoptosis in proliferating lymphocytes and reducing the release of pro-inflammatory cytokines such as interleukin-17 and interferon-γ. Vidofludimus calcium exhibits partial activity as an FXR agonist. It is currently being evaluated in clinical trials (NCT04595825) as a potential treatment for PSC. The results from a pilot study in 18 patients demonstrated a normalization of ALT in approx. 30% of patients after 24 weeks [157]. Positive results for autologous regulatory T-cell transfer in refractory ulcerative colitis with concomitant primary sclerosing cholangitis have been also reported, suggesting that adoptive Treg therapy might be effective in refractory UC and might open new avenues for clinical trials in PSC [158].

### 8.10. Microbiota-Based Therapies in PSC

Targeting the gut microbiota as a therapeutic strategy for PSC presents a compelling area for exploration [108,159].

### 8.11. Antibiotic Therapy

Multiple antibiotics, including metronidazole, azithromycin, vancomycin, and minocycline, have been investigated as therapies for PSC, with some showing promising effects in small studies [159]. The most widely investigated antibiotic is vancomycin. There are two randomized controlled trials investigating vancomycin as therapy for PSC. In one of these trials [160], 33 adult PSC patients, 21 with concomitant IBD, on UDCA treatment were administered vancomycin 125 mg four times a day (*n* = 18) or placebo (*n* = 11) for 12 weeks. The prespecified end point, a reduction in Mayo Risk Score, was achieved in the vancomycin group, but not in the placebo group, which also failed to reduce ALP. Patients in the vancomycin group experienced a significant reduction in subjective symptoms, including fatigue and itching. In a second study [112], 35 adult patients with PSC were randomized to receive 125 mg or 250 mg vancomycin four times a day, or 250 mg or 500 mg metronidazole three times a day, for 12 weeks. The primary end point (a decrease in the serum ALP at 12 weeks) was met by both the low- and high-dose vancomycin groups, but not by the metronidazole groups. The Mayo Risk Score was significantly reduced in the low-dose vancomycin group (−0.55, *p* = 0.02), but no other secondary biochemical endpoints were reached. Despite some promising outcomes, the group sizes in this study were small, and it was not powered to detect small changes. In summary, while some studies have shown promising results, including improvements in liver function tests and decreased expression/production of inflammatory biomarkers, clinical evidence remains limited, and vancomycin is not yet an approved treatment for PSC. Finally, the risk of antibiotic resistance development be considered. More robust clinical trials are needed. There are at least two mechanisms that might support o role for vancomycin in the treatment of PSC: i.e., microbiome modulation and immunomodulation, since some studies suggest that vancomycin may alter T-cell activity or reduce pro-inflammatory cytokines, contributing to its therapeutic effect in PSC.

### 8.12. Fecal Microbiota Transplantation (FMT)

To date, one study has reported the effects of FMT in PSC [161]. This was an open-label pilot study carried out in 10 patients with PSC with concurrent IBD and ALP 1.5× the upper limit of normal. The patients underwent a single FMT by colonoscopy. The results demonstrated that FMT in PSC is safe. In addition, increases in bacterial diversity and engraftment were found to correlate with an improvement in ALP.

### 8.13. Treating Pruritus

Various agents have been used to treat itching in PSC patients, including rifaximin; a PXR agonist; and bile acid sequestrants such as cholestyramine, naltrelone, sertraline, and fenofibrates. All these agents but fibrates have shown limited efficacy. In the Fibrates for Itch (FITCH) in Fibrosing Cholangiopathies study [8], 70 patients with PBC and PSC (44 patients) were randomized to receive placebo or bezafibrate. The primary end point of the study was a ≥50% reduction in pruritus (VAS; intention-to-treat), which was met by bezafibrate in 45% of patients (41% PSC, 55% PBC) and by 11% of patients in the placebo arm (*p* = 0.003 vs. bezafibrate). Other approaches that are currently being investigated involve MAS-related G-protein-coupled receptors (MRGPR) antagonists, particularly [162] and MrgprC11 [163] antagonists. These receptors mediate histamine-independent pruritus and are promising therapeutic targets for antihistamine-resistant pruritus, typically in PBC and PSC [164]. MRGPRs are primarily expressed in sensory neurons of the dorsal root ganglia (DRG) and trigeminal ganglia, and are involved in detecting and transmitting itch signals from the skin to the central nervous system, including bilirubin and bile acids [8]. A clinical trial with an MRGPRS antagonist, EP547, called the Randomized, Double-Blind, Placebo-Controlled Study to Evaluate the Effects of EP547 in Subjects with Cholestatic Pruritus Due to Primary Biliary Cholangitis or Primary Sclerosing Cholangitis, NCT05525520, is currently ongoing.

## 9. Conclusions

Despite much progress, therapy for PSC remains suboptimal, as is our understanding of its pathophysiology and the development of effective medical treatments. At present, liver transplantation remains the sole curative treatment for patients with end-stage liver disease or those experiencing severe complications such as intractable pruritus or recurrent cholangitis. Current data show that patients receiving deceased donor allografts have high 5-year survival rates of ≈85%. Localized cholangiocarcinoma or high-grade biliary dysplasia may also be considered for transplantation. Furthermore, while there is a risk for PSC recurrence in approx. ≈ 20% of transplanted grafts, this recurrence is generally benign. Survival rates are comparable to those without recurrence, underscoring the effectiveness of liver transplantation in managing advanced PSC despite the risk of graft recurrence.

Medical therapy, in contrast, remains suboptimal. UDCA should be offered to patients with PSC at a standard dose up to 15 mg/kg [165], but, in contrast to PBC, its efficacy is currently limited to improving liver biochemistry at moderate doses, with some potential chemoprotective benefits for patients with PSC-IBD.

Advances in manipulating the bile acid pathways using nuclear receptor agonists such as FXR have proven difficult, because the additional repression of bile acid synthesis caused by FXR agonism could be detrimental to PSC patients. ASTB/IBAP inhibitors to prevent bile acid absorption are promising. Additionally, other bile acid-based therapies that are currently under development are GPBAR1 agonists for promoting cholangiocytes and immune homeostasis in the liver and intestine, which might be of utility in PSC-IBD. The MRGPR antagonist for treating pruritus also targets a specific bile acid-regulated receptor, MRGPRX4 [162,163], that is activated by bile acids and bilirubin. PPARs, including the recently approved selective PPARs agonists elafibranor and seladelpar [166,167], that have proven effective in PBC patients are also currently being investigated in PSC.

Microbiome modulation offers promising opportunities, especially for PSC patients with concurrent IBD, but there is a need for large trials to support this approach.

Additionally, the development of novel radiologic scoring systems and liver fibrosis stratification will help to tailor therapy to specific patient subsets.

## Figures and Tables

**Figure 1 cells-13-01650-f001:**
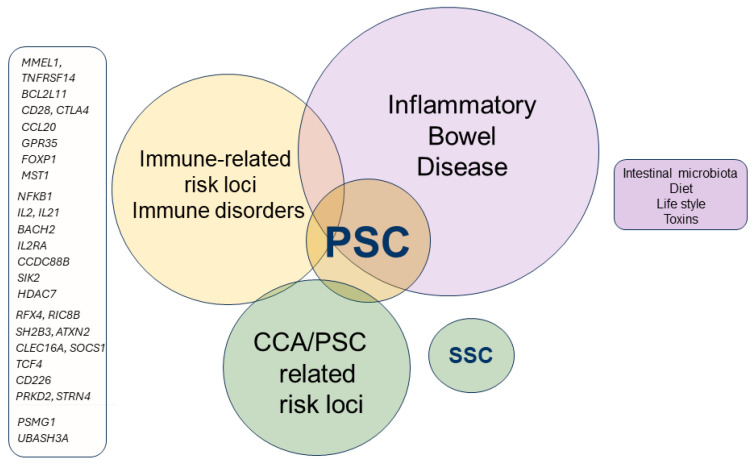
Risk factors involved in development of PSC. PSC: primary sclerosing cholangitis; SSC: secondary sclerosing cholangitis; CCA/PSC: cholangiocarcinoma in PSC.

**Figure 2 cells-13-01650-f002:**
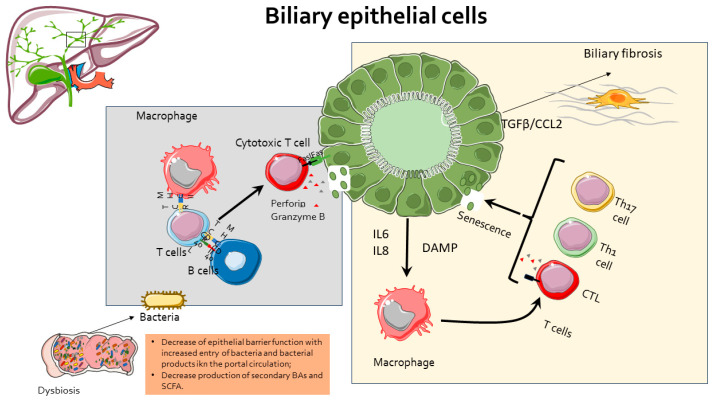
Role of cholangiocytes in PSC pathogenesis. DAMP: damage-associated molecular patterns. TGF, transforming growth factor beta. CCL2, chemokine (C-C motif) ligand 2.

**Figure 3 cells-13-01650-f003:**
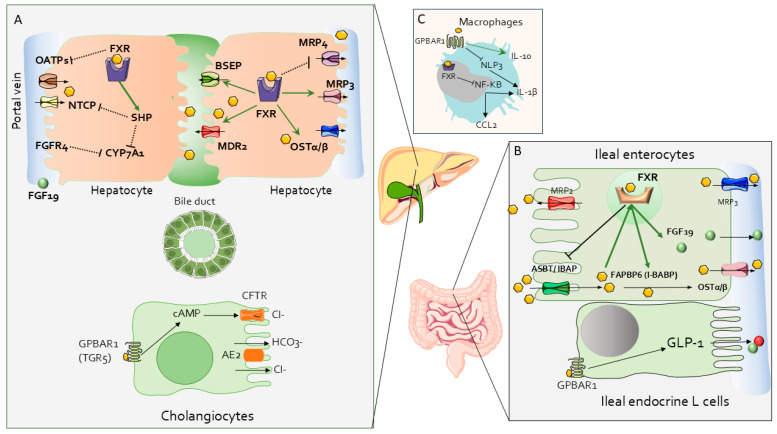
FXR- and GPBAR1-activated pathways in hepatic (**A**), intestinal (**B**), and immune cells (**C**). Pathway induction is indicated by green arrows; pathway inhibition by black dotted lines. Abbreviations: AE2: Anion Exchange protein 2; ASBT/IBAP: Apical Sodium–Bile Acid Transporter/Ileal Bile Acid Transporter; BSEP: Bile Salt Export Pump; cAMP: cyclic Adenosine Monophosphate; CCL2: C-C Motif Chemokine Ligand 2; CFTR: Cystic Fibrosis Transmembrane Regulator; Cl-: Chloride Ion; CYP7A1: Cholesterol-7-alpha-hydroxylase; FAPBP6 (I-BABP): Fatty Acid Binding Protein 6 (Ileal-Bile Acid Binding Protein); FGF19: Fibroblast Growth Factor 19; FGFR4: Fibroblast Growth Factor Receptor 4; FXR: farnesoid X receptor; GLP-1: Glucagon-like Peptide 1; GPBAR1 (TGR5): G Protein-Coupled Bile Acid Receptor (Takeda G protein-coupled Receptor 5); HCO3-: Bicarbonate Ion; IL-1 β: Interleukin 1 beta; IL-10: Interleukin 10; MDR2: multidrug resistance protein 2; MRP3: Multidrug Resistance-associated Protein 3; MRP4: multidrug resistance-associated Protein 4; NF-κB: Nuclear Factor kappa B; NLRP3: NOD-, LRR- and Pyrin domain-containing protein 3; OATPs: Organic Anion-Transporting Polypeptides; OSTα/β: Organic Solute Transporter alpha/beta.

**Figure 4 cells-13-01650-f004:**
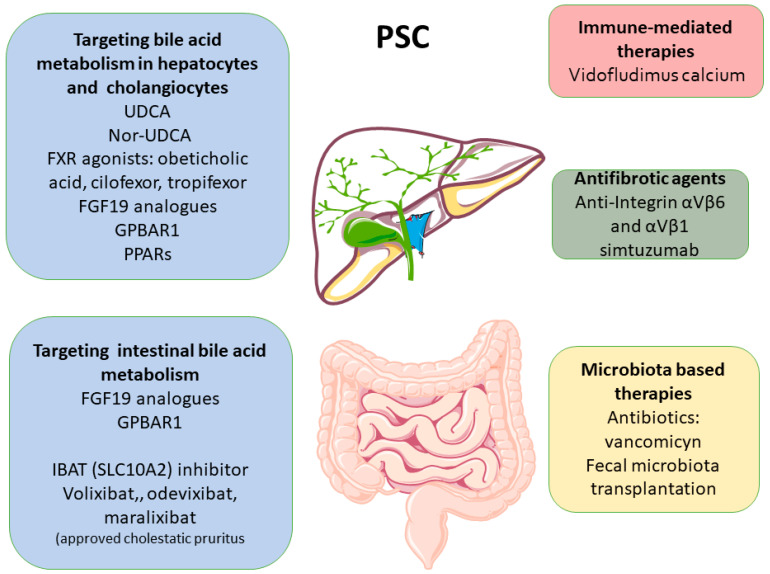
Evolving approaches to PSC.

**Table 1 cells-13-01650-t001:** Clinical variants of PSC and secondary sclerosing cholangitis.

Type	Main Histopathology and Genetic Characteristics
Classical, large-duct PSC	Is the prototypical form of PSC, involves large extrahepatic bile ducts, and is more prevalent in males, with a ratio of 3:2. Age of onset is the fourth decade. Associated with IBD, generally pancolitis, in 60–70% of cases. Progression to transplant ranges from 12 to 20 years. Associated with HLA haplotypes: HLA-A*01; HLA-B*08 or HLA-DRB1*03031.
Non-IBD PSC	Equally distributed among males and females. Onset at older age. Same HLA haplotypes as PSC-IBD. Better outcome.
Small-bile-duct PSC	This form accounts for 10% of PSC. This phenotype lacks typical magnetic resonance cholangiopancreatography (MRCP) characteristics. IBD is required for diagnosis. Patients might be considered PBC-AMA. If IBD is not present, the HLA haplotypes differ from the large-duct PSC.
Overlap syndrome: PSC-autoimmune hepatitis	A variable percentage of PSC patients show features of auto-immune hepatitis, showing pANCA, ANA and SMA antibodies. Typically shows elevated levels of AST and ALT (up to ten folds) at onset.
Pediatric PSC	PSC in children shares most of the features of large-duct PSC, including association with IBD. Association with autoimmune hepatitis occurs in 10% of these patients.
Non-Caucasian PSC	PSC occurs in the African-American population with the same prevalence as the Caucasian population. The prevalent haplotype is HLA-B*08. Less frequent association with IBD.
Secondary sclerosing cholangitis	Infections, IgG4-associated SSC, HIV infection, sarcoidosis, biliary stones, traumatic, ischemic. Cholangiocarcinoma, pancreatic and ampullary cancer. Papillary stenosis. Chronic pancreatitis. Infestation (ascaris, liver fluke).

**Table 2 cells-13-01650-t002:** Currently ongoing clinical trials on PSC therapeutics.

Study ID	Status	Drug-Target	CT Phase	Primary and Secondary End Points and Results
**NCT04480840**	Active, not recruiting	PLN-74809, αvβ6 and αvβ1 inhibitor	2	Evaluation of safety, tolerability, and PK of PLN-74809 in participants with PSC and suspected liver fibrosis
**NCT05642468**	Recruiting	A3907, ASBT inhibitor	2	Evaluation of safety and tolerability of A3907 in PSC patients
**NCT04663308**	Recruiting	Volixibat, ASBT inhibitor	2	Evaluation of Volixibat use in PSC-associated itching treatment Secondary Evaluation of Volixibat impact on PSC disease progression
**NCT05896137**	Recruiting	CS0159, FXR inhibitor	2	Evaluation of safety, tolerability, and efficacy of CS0159 in patients with PSC
**NCT05082779**	Completed	CS0159, FXR inhibitor	1	**Primary** Evaluation of safety, tolerability, and PK and PD profiles of CS0159 in SAD and MAD studies **Secondary** Assessment of Food Effect (FE) on single-dose PK profile of CS0159 from SAD study in healthy subjects
**NCT05295680**	Recruiting	Hymecromone, antineoplastic agent/hyaluronic acid synthesis inhibitor	2	**Primary** Evaluation of the efficacy of hymecromone plus standard of care (SOC) compared with SOC alone in the treatment of adolescents and adults with primary sclerosing cholangitis (PSC) **Secondary** Evaluation of changes in ALP from baseline to 6 months post-treatment following treatment with hymecromone plus SOC compared with SOC Evaluation of changes in PSC biomarkers: fibrotic effect (FibroScan), inflammation (serum Hyaluronan, HA), T-cell count
**NCT05866809**	Completed	HK-660S, anti-inflammatory/anti-fibrotic agent	2	Evaluation of improvement of bile duct strictures via MRCP and ALP levels assessment following the administration of HK-660S in patients with PSC
**NCT03722576**	Completed with results	Vidofludimus	2	**Primary** Evaluation of safety, tolerability, and efficacy of daily dosing with VC over a 6-month period via assessment of ALP and liver biochemistry (3 and 6 months). **Secondary** Assessment of IL-17 and IFNγ (6 weeks and 6 months) **Results** - Combined reduction in AST and ALT: 27.3% for Abnormal AST and ALT: 80% - Abnormal total bilirubin: 30%; abnormal direct bilirubin: 40%
**NCT05627362**	Recruiting	Elafibranor, PPARα/δ agonist	2	**Primary** Evaluation of safety and side effects of Elafibranor in participants with PSC **Secondary** Evaluation of drug’s effects on blood tests and other tests related to PSC disease activity
**NCT04024813**	Completed	Seladelpar, PPARδ agonist	2	Evaluation efficacy, safety, and tolerability of effects of Seladelpar compared to placebo in patients with PSC
**NCT04309773**	Recruiting	Benzafibrate, PPARα agonist	3	**Primary** Evaluation of effects of Benzafibrate treatment compared to placebo on efficacy and safety in patients with PSC despite standard UDCA therapy **Secondary** Effects of Benzafibrate treatment compared to placebo on persistent cholestasis in patients with PSC despite standard UDCA therapy
**NCT06026865**	Recruiting	S-adenosyl-methionine (SAMe),	N.A.	**Primary** Investigation of clinical effects (liver biochemistry, health-related quality of life, liver stiffness) of SAMe in patients with PSC **Secondary** Investigation of underlying mechanisms of hepatoprotection (plasma concentrations of SOD2, FGF-19, TNF-α, IL-6, and others) of SAMe in patients with PSC
**NCT05525520**	Recruiting	EP547, MrgprX4 antagonist	2	Evaluation of the effects of EP547 in subjects with cholestatic pruritus due to PBC or PSC
**NCT04060147**	Terminated with results	Cilofexor, FXR agonist	1	Evaluation of safety and tolerability of escalating doses of CILO in participants with PSC and compensated cirrhosis **Results** - Treatment-emergent adverse events (TEAEs): 81.8% - Treatment-emergent serious adverse events (SAEs): 0%
**NCT03890120**	Terminated with results	Cilofexor, FXR agonist	3	Cilofexor reduce the risk of fibrosis progression among non-cirrhotic adults with PSC **Results** - Liver fibrosis progression in blinded phase, cilofexor vs. placebo: 30.8% vs. 32.8% - Severe adverse event (SAEs) in blinded phase, cilofexor vs. placebo: 19.1% vs. 18.7% - Changes in serum ALT concentration in blinded phase, cilofdexor vs. placebo (LSM): −13 vs. −3 - Changes in serum fasting total BAs concentration in blinded phase, cilofexor vs placebo (LSM): 7.2 vs. 9.8 - Liver fibrosis improvement in blinded phase, cilofexor vs. placebo: 25.6% vs. 17.2%
**NCT05912387**	Recruiting	RosuvastatinHMG-CoA reductase inhibitor	Early 1	Evaluation of bile acids profile (total BAs, SBAs:PBAs ratio, conjugated:deconjugated BAs ratio, and other) and microbiome impact of Rosuvastatin in PSC patients
**NCT04133792**	Recruiting	Simvastatin	3	Assessment of effect of PSC prognosis according to long-term intake of Simvastatin (death, liver transplantation, cholangiocarcinoma, esophageal varices bleeding)
**NCT05876182**	Recruiting	Vancomyci,	2	Evaluation of safety and efficacy of two doses of oral Vancomycin in subjects between 15 and 70 years old with PSC
**NCT03710122**	Recruiting	Vancomycin	2/3	**Primary** Evaluation of safety and efficacy of Vancomycin for therapy in adult patients with PSC via assessment of serum ALP levels, liver fibrosis, and pro-inflammatory cytokines (TGF-ß, IL-4 and others) **Secondary** Determination of changes in the intestinal microbiota in relation to the use of Vancomycin and its correlation with changes in serum ALP and liver fibrosis
**NCT06037577**	Completed	CM-101, anti-CCl24 monoclonal antibody	1	Evaluation of safety, tolerability, PKs, and PDs of single escalating subcutaneous doses of CM-101 in healthy male subjects
**NCT04595825**	Active, not recruiting	CM-101, anti-CCl24 monoclonal antibody	2	Evaluation of safety, tolerability, and activity of the anti-human CCL24 monoclonal antibody CM-101 in adult subjects with PSC
**NCT06553768**	Not yet recruiting	Maralixibat, IBAT inhibitor	3	**Primary** Evaluation of the mean change in the ItchRO(Obs) severity score from baseline to average of week 13 to week 20 **Secondary** Evaluation of the mean change in total sBA levels from baseline to average of week 12 and week 20
**NCT06286709**	Recruiting	Fecal Microbiota Transplantation	2	**Primary** Evaluation of reduction in serum ALP measured at 48 weeks following the first dose of FMT or FMT placebo **Secondary** Evaluation of PSC-PRO, SF-36, 5D-Itch, and SIBDQ questionnaires at screening (week-2) and weeks 8, 12, 24, 36, and 48; evaluation of VCTE at screening and week 48; evaluation of ELF at weeks 1, 5, 12, and 48; evaluation of AST, ALT, bilirubin, gGT, and albumin and assessment of UK-PSC score and Amsterdam-Oxford PSC score at screening and weeks 1, 3, 5, 8, 12, 24, 36, and 48; quantitative assessment of SES-CD and fecal calprotectin at screening and weeks 1, 5, 12, and 48; evaluation of occurrence of adverse events measured by CTCAE v5.0
**NCT06197308**	Recruiting	Vancomycin	Early 1	Evaluation of safety and feasibility of microbiota transplant therapy (MTT) by determination of the frequency of serious adverse events or other adverse events and determination of the proportion of subjects taking 100% of the MTT per protocol, respectively
**NCT06095986**	Not yet recruiting	Aramchol meglumine, Stearoyl-CoA desaturase 1 inhibitor	2	**Primary** Evaluation of changes in ALP levels from baseline to week 48 **Secondary** Evaluation of changes in Nakanuma stage classification, ELF test, MRCP, Gadoxetate clearance test, 5D-Itch scale, PROMIS-19 questionnaire, and Mayo IBD symptom severity score, as well as rMRS, UK-PSC, and PREsTo, from baseline to 48 weeks

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
