# Peer review of "Bile Acids-Based Therapies for Primary Sclerosing Cholangitis: Current Landscape and Future Developments"

_cells, 2024, doi:10.3390/cells13191650_

Round 1

Reviewer 1 Report

Comments and Suggestions for Authors

This is a comprehensive review manuscript for PSC. Some inadequate statements regarding bile acid metabolism need to be corrected.

1.     Line 204. "within the" what?

2.     Lines 231-232. Conjugated BAs are actively secreted into the canalicular space by BSEP and MRP2 (not MDR3).

3.     Figure 3A and lines 310-311. MDR2 is a rodent ortholog of human MDR3. Does this figure show the human liver or mice liver? Does FXR inhibit only NTCP or also OATPs? The effects of SHP and FGFR4 on CYP7A1 are not depicted.

Author Response

  1. Line 204. "within the" what?

R1. Done, see new  sentence line 203 (in yellow).

  1. Lines 231-232. Conjugated BAs are actively secreted into the canalicular space by BSEP and MRP2 (not MDR3).

R2. Amended.

  1. Figure 3A and lines 310-311. MDR2 is a rodent ortholog of human MDR3. Does this figure show the human liver or mice liver? Does FXR inhibit only NTCP or also OATPs? The effects of SHP and FGFR4 on CYP7A1 are not depicted.

R3. Amended. See legend to Figure 3.

Reviewer 2 Report

Comments and Suggestions for Authors

This is a comprehensive well referenced review of PSC.  My only major issue is that there is a 3 page discussion of bile acid metabolism, nucler recepetors and FXR (section 5 through 5.1) that  has nothing to do with PSC and could easily be removed as I found it interfering, and somewhat gratuitous. It is followed by a section that does deal with bile acids and PSC which is fine. Otherwise I had a number of minor issues that I append here:

Comments to Authors:

P 5,line 170 . The Lab Investigation paper does not compare PSC cholangiocyte cultures with normal cholangiocytes, rather to a NHC cell line . Thus the statement is not entirely correct and might be modified re: abnormal T.J.s.

p.6. line 215, “tun” should be “turn”.

p.6 , line 231, MDR3 is not a bile acid transporter on the apical membrane., it is a phosphatidylcholine transporter.  MRP3  on the basolateral mmbrane is up regulated in cholestasis and can efflux bile acid glucuronides conjugates.

p. 7 ,, line 254: “t” is missing.

p.7. Section 5 to 5.1.1  is a n extensive general discussion of bile acid metabolism , nuclear receptors and FXR without reference to PSC, As such it is unnecessary and distracting and could be removed. It is followed by an appropriate discussion of bile acids and PSC.

p. 11, section 7.1: The section on Biomarkers should  also include discussion of Fib-4, ELF ,procollagen although mentioned later to some extent.

p.12, line 502: ref 89 seems wrong.

P,13, line 539.  Authors should refer to the Lindor paper which describes the AASLD PSC guidelines , rather then reference a paper that refers to them.

p.13, line 557.  Are references 129 and 130 correct?

p.13, section 8.3:: Authors should state more clearly their opinioin re: use of OCA in PSC given that the studies did not provide an evidence of efficacy and side effects were significant.

Section 8.5:  Reference 149 refers to TGR5.  This is the same as GPBAR1 and should be introduced when first mentioning this receptor for clarity.

p.14, line 628: This reference is 15 years old so I might remove the term “recent”.

p.16, line 887: LAT  should be SGPT?  Or ALT?

p.16, line 708: This sentence needs to be clearer as to whether just metronidazole in contrast to vancomycin was effective.  Authors might also mention that antibiotic resistance is a concern re: long term therapy with vancomycin.

Section 8.13, line 732: “meet” should be “met”.  Line 733,, “harm” should be “arm”.

Section 9.  Conclusion. Line 770: Elafibranor and Seladelpar are “selective” PPAR Agonists, not “pan” PPAR agonists like Bezafibrate.

Author Response

This is a comprehensive well referenced review of PSC.  My only major issue is that there is a 3 page discussion of bile acid metabolism, nucler recepetors and FXR (section 5 through 5.1) that  has nothing to do with PSC and could easily be removed as I found it interfering, and somewhat gratuitous. It is followed by a section that does deal with bile acids and PSC which is fine. Otherwise I had a number of minor issues that I append here:

Comments to Authors:

1. p.5 line 170. The Lab Investigation paper does not compare PSC cholangiocyte cultures with normal cholangiocytes, rather to a NHC cell line. Thus the statement is not entirely correct and might be modified re: abnormal T.J.s.Thanks.

R1. Done as suggested.

2. p.6 line 215, “tun” should be “turn”.

R2. Done.

3. p.6 line 231, MDR3 is not a bile acid transporter on the apical membrane., it is a phosphatidylcholine transporter.  MRP3  on the basolateral mmbrane is up regulated in cholestasis and can efflux bile acid glucuronides conjugates.

R3. Done. See revised Figure 3. MRP2 removed.

4. p.7 line 254: “t” is missing.

R4. Not found.

5. p.7, Section 5 to 5.1.1 is a n extensive general discussion of bile acid metabolism , nuclear receptors and FXR without reference to PSC, As such it is unnecessary and distracting and could be removed. It is followed by an appropriate discussion of bile acids and PSC.

R5. Thanks for the suggestion, however this paragraph deals with the general mechanism of synthesis and excretion and intestinal metabolism of bile acids and we believe it is necessary to introduce the  concept of dysregulated bile acid metabolism in PSC.

6. section 7.1: The section on Biomarkers should  also include discussion of Fib-4, ELF ,procollagen although mentioned later to some extent.

R6. p.11, Paragraph 7.3 deals with the fibrosis biomarkers, and is not incorporated in validated models such as the PReSTO or other discussed in Paragraph 7.1 like the ELF score I mentioned in paragraph 7.3. We have now also mentioned the Fib-4. See ref. 119.

7. p.12, line 502: ref 89 seems wrong.

R7. Reference removed.

8. p.13, line 539.  Authors should refer to the Lindor paper which describes the AASLD PSC guidelines , rather then reference a paper that refers to them.

R8. Amended to reference 33 the most recent guidelines form AASLD.

9. p.13, line 557.  Are references 129 and 130 correct?

R9. Reference 130 is the original paper form Fiorucci, showing the anti-fibrotic effects of FXR, and included INT747 (the original name of obeticholic acid). Reference 129 has been removed.

10. p.13, section 8.3: Authors should state more clearly their opinioin re: use of OCA in PSC given that the studies did not provide an evidence of efficacy and side effects were significant.

R10. Done. See "The severity of these side effects will likely preclude the further development of OCA in PSC. Furthermore, the European Medicines Agency has recently recommended the revoking of conditional marketing authorization of OCA in PBC treatment (June 28, 2024- https://www.ema.europa.eu/en/news/ema-recommends-revoking-conditional-marketing-authorisation-ocaliva).”

11. Section 8.5:  Reference 149 refers to TGR5.  This is the same as GPBAR1 and should be introduced when first mentioning this receptor for clarity.

R11. Clarified.

12. p.14, line 628: This reference is 15 years old so I might remove the term “recent”.

R12. Done.

13. p.16, line 887: LAT  should be SGPT?  Or ALT?

R13. LAT changed to ALT.

14. p.16, line 708: This sentence needs to be clearer as to whether just metronidazole in contrast to vancomycin was effective.  Authors might also mention that antibiotic resistance is a concern re: long term therapy with vancomycin.

R14. As far as antibiotic resistance is considered, various studies have confirmed that this is not the case was used for 12 weeks in patients with PSC. However, we had a sentence.

15. Section 8.13, line 732: “meet” should be “met”.  Line 733,, “harm” should be “arm”.

R15. Done.

16. Section 9. Conclusion. Line 770: Elafibranor and Seladelpar are “selective” PPAR Agonists, not “pan” PPAR agonists like Bezafibrate.

R16. Done.